# Assessment of the performance and challenges in the implementation of the test, treat and track (T3) strategy for malaria control among children under-five years in Ghana

Margaret Kweku, Joyce B. Der◉*, William K. Blankson, Haruna M. Salisu◉, Francis Arizie, Sorengmen A. Ziema, Jonathan M. Gmanyami, Fortress Y. Aku, Martin Adjuik

School of Public Health, University of Health and Allied Sciences, Hohoe, Volta Region, Ghana

* jdberkumwine@uhas.edu.gh

**Data Availability Statement:** All relevant data are within the manuscript and its Supporting information files.

## Abstract

### Background

The World Health Organization recommended the Test, Treat and Track (T3) strategy for malaria control that, every suspected malaria case should be tested prior to treatment with Artemisinin-based combination therapy (ACT) and tracked. We assessed the performance and challenges in the implementation of T3 strategy among children under-five years in Volta and Oti Regions of Ghana.

### Method

A descriptive cross-sectional study was carried in 69 health facilities. Exit interviews were conducted for caregivers of children with fever using a semi-structured questionnaire. Clinicians were interviewed at the out-patient department in each facility. Descriptive statistics was conducted, Chi-square test and logistic regression were used to determine the associations between completion of T3 and independent variables.

### Results

Most children, 818/900 (90.9%) were tested for malaria and 600/818 (73.4%) were positive for malaria parasitaemia using rapid diagnostic test. Of those testing positive for malaria, 530/600 (88.3%) received treatment with ACTs. Half, (109/218) of the children testing negative for malaria also received ACTs. Also, 67/82 (81.7%) of children not tested for malaria received ACTs. Only 408/900 (45.3%) children completed T3 with Community Health-based Planning Services (CHPS) compound having the highest completion rate 202/314 (64.3%). CHPS Compounds were 6.55 times more likely to complete T3 compared to the hospitals [(95% CI: 3.77, 11.35), p<0.001]. Health facilities with laboratory services were

**Funding:** The author(s) received no specific funding for this work.

**Competing interests:** The authors have declared that no competing interests exist.

2.08 times more likely to complete T3 [(95% CI: 1.55, 2.79), p<0.001] The main challenge identified was clinicians' perception that RDTs do not give accurate results.

## Conclusion

Testing fever cases for malaria before treatment and treating positive cases with ACTs was high. Treating negative cases and those not tested with ACTs was also high. Health facilities having laboratory services and facility being CHPS compounds were key predictors of completing T3. Clinician's not trusting RDT results can affect the T3 strategy in malaria control. Periodic training/monitoring is required to sustain adherence to the strategy.

## Introduction

Malaria is a complex disease that varies widely in epidemiology and clinical manifestations and remains the leading cause of death in the world [1]. Malaria diagnosis over the years was mainly based on signs and symptoms. Thus, malaria was mostly treated presumptively which led to overuse of anti-malarial drugs, unaccountability of resources and unnecessary suffering especially where fever was associated with other infections [2]. Malaria is still endemic in 91 nations and over 90% of all malaria deaths currently occur in sub-Saharan Africa (SSA) [3]. Two nations–the Democratic Republic of the Congo and Nigeria account for about 40% of global malaria mortality [4]. The World Health Organization (WHO) has recommended several interventions which include the use of Long-Lasting Insecticide Treated bed Nets (LLIN), Indoor Residual Spraying (IRS) and Larval control as well as chemoprophylaxis and the use of Artemisinin-based combination therapy (ACT) as first line treatment for uncomplicated malaria in malaria endemic countries for the prevention and control of malaria over the past decades [5].

On World Malaria Day 2012, WHO Director-General Margaret Chan launched a new malaria control initiative called Test, Treat and Track (T3) [6]. The initiative seeks to focus the attention of policy-makers and donors on the importance of adopting WHO's latest evidence-based recommendations on diagnostic testing, treatment and surveillance, and on updating existing malaria control and elimination strategies, as well as country-specific operational plans [7]. The main rational behind the implementation of T3 is that before malaria can be eliminated, there is the need to ensure every suspected malaria case is tested, every confirmed case is treated with a quality-assured antimalarial medicine and that the disease is tracked through timely and accurate surveillance systems [6].

The T3 strategy for malaria control recommends that every suspected malaria case should be tested prior to treatment. The testing could either be done by using rapid diagnostic test (RDT) or microscopy, and all diagnostic tools for malaria cases must be of quality with enough trained personnel across all levels of the health system to provide the service [2]. This is aimed at improving the quality of care and ensuring that antimalarial medicines are used rationally and correctly [2]. In Ghana, treatment of malaria is according to the national treatment guidelines where the first line antimalarials for uncomplicated malaria are Artesunate-Amodiaquine (AS-AQ) and Artemether-Lumefantrine (AL). Alternative ACTs or second line antimalarial for those unable to tolerate AS-AQ is Artemether-Lumefantrine (AL) or DihydroArtemisinin Piperaquine (DHAP). Severe malaria is treated with parenteral Artesunate (intravenous (IV)/ Intramuscular (IM)) initially then revert to oral treatment as soon as patient's condition permits. Treatment of malaria is provided with the necessary support therapy such as antipyretics

for fever for uncomplicated malaria and IV fluids for severe malaria as and when appropriate [8].

RDTs for malaria provide an opportunity for improved point-of-care diagnosis and better disease management in malaria-endemic countries [9]. Nevertheless, fever cases are mostly equated to malaria and treated as such in some health facilities without parasitological diagnosis. In 2014, Baiden et al., revealed that the probability of fever that could be attributed to malaria was as high as 61% and 67% in Ghana and Kenya respectively [10]. Studies have shown that children under 5 years are the mostly the ones not tested before treatment [11].

Recent studies in Ghana have shown that a substantial proportion of febrile cases are still presumptively treated as malaria despite the availability of RDTs and the emphasis on testing and treatment based on test results with proportions ranging between 58.5%–91.2% [12, 13]. The cost-effectiveness of implementing test-based management of malaria hinges on health workers adhering to test-results and restricting ACTs to confirmed cases while looking for other causes of fever in the test-negative cases [10]. Similarly, the level of clinician's adherence to the test of fever cases, negative test results, and tracking of malaria cases through a timely surveillance system are major problems that need attention [13].

A good performance of the T3 strategy for malaria is measured by being tested, treated with an antimalarial and asked to come for review. Thus, all the 3 Ts must be in place. Missing a "T" means the strategy was not well adhered to. Challenges faced by clinicians in implementing the T3 strategy include frequent RDT stock out and lack of diagnostic facilities as reported by studies in Ghana [12, 13]. This study assessed the performance of the T3 strategy implementation in the Volta and Oti regions. It determined the proportion of suspected malaria cases tested by clinicians prior to treatment and proportion of confirmed malaria cases that received treatment according to the T3 malaria policy. It also determined the challenges of the implementation of T3 strategy.

## Materials and method

### Study site description

The study was carried out in the Volta region prior to the region being split into Volta and Oti regions in 2018. The regions are two out of the sixteen administrative regions in Ghana with a total of 26 administrative municipalities/districts as shown in Fig 1. The regions stretch across all the ecological zones of the country namely: Coastal Savannah, Middle Forest and Northern/Sahel Savannah [14].

The Volta and Oti regions have a total of 698 health facilities including 28 hospitals, 4 polyclinics, 154 health centres, 44 clinics, 14 maternity homes and 454 Community-Based Health Planning and Services (CHPS) compounds. The two regions recorded 2,294,337 outpatient attendance in 2017 [15].

In 2016, the region had the third highest (28%) malaria prevalence and the highest case fatality indicators in the region. The 2015 Volta regional prevalence of malaria among children aged 6–59 months were 36.6% and 25.2% for RDT and microscopy respectively [16]. There was an increment to 36.9% and 27.3% for RDT and microscopy respectively in 2017 [17].

### Study population

The study had two different populations: (i) children under five years with their caregivers who reported to the health facilities with fever or history of fever and were managed by clinicians and (ii) clinicians rendering services at the various health facilities during the time of the study.

**Inclusion and exclusion criteria for children.** Children aged less than 5 years who attended the Out-Patient Departments (OPDs) of selected health facilities with fever or history

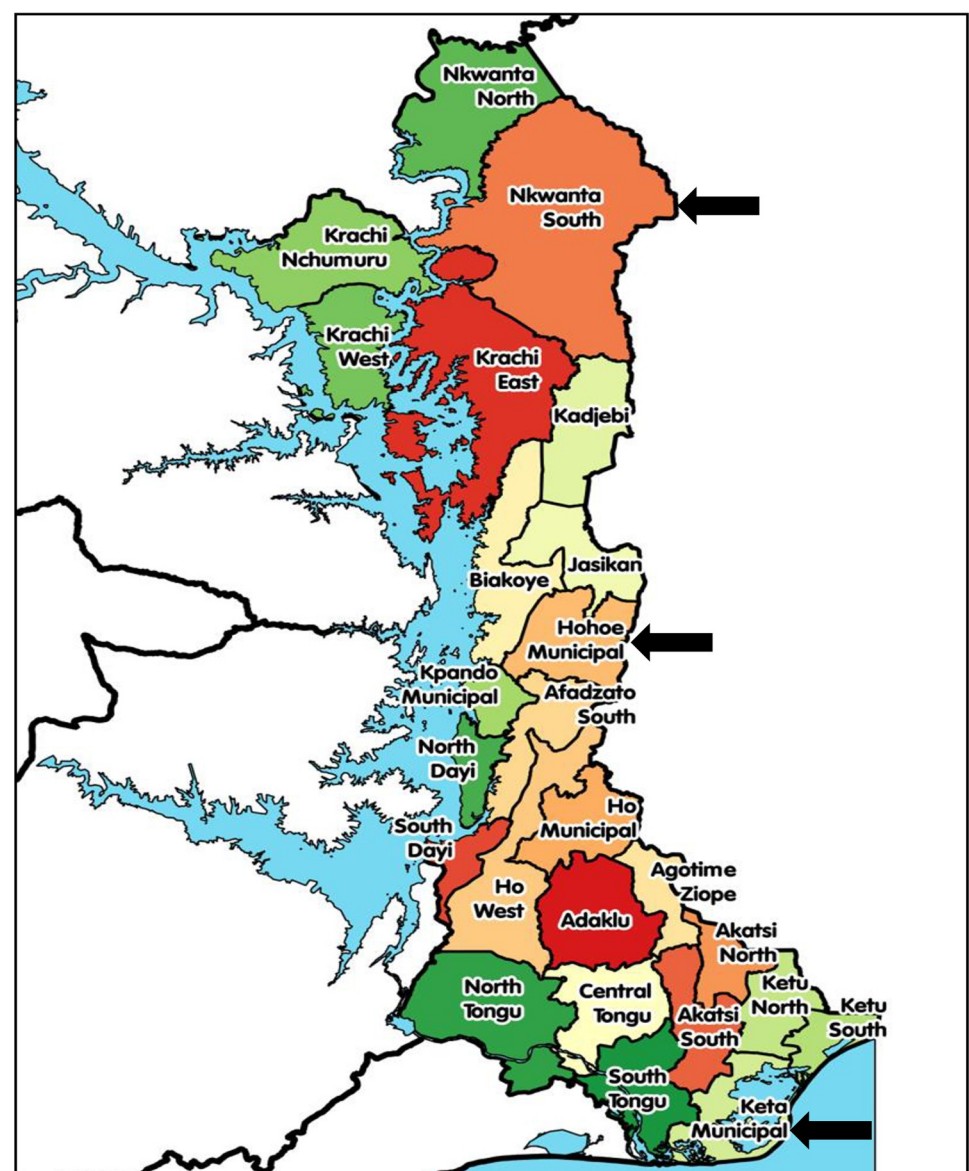

**Fig 1. Map showing the three study municipalities/districts in the Volta and Oti regions.**

of fever and whose parent/guardian consented to participate in the study were included. Children who were seriously ill requiring hospital admission were excluded.

**Inclusion and exclusion criteria for clinicians.** All clinicians who were assigned the duties of prescribing during the study period, were available and consented to participate were included in the study.

## Study design

A descriptive cross-sectional study design was used for the study. From September to October 2017, data were collected from caregivers who visited any of the selected health facilities with their children presenting with fever or were diagnosed with malaria. Data were collected as they exited the health facility after they had been attended to by the clinicians. Information

collected from the caregivers included whether the child was treated for malaria, whether the child was tested before treatment for malaria and whether they were asked to return for follow-up and when they were supposed to return. Clinicians at the health facilities were also interviewed to determine the factors that prevented them from adhering to the T3 strategy. All data were collected using a pre-tested semi-structured questionnaire.

## Sample size determination

For the sample of children, according to Agandaa et al., [12], 42.5% (P) is the proportion of completion of Test, Treat and Track strategy for malaria case management in children under 5 years old in Bongo District, Ghana.

Using Open Epi version 3 Source calculator [18], the sample size was calculated as follows;

$$= \frac{Deff * N * p * (1 - p)}{d^2 \big/ z^2 * (N - 1) + p * (1 - p)} \quad \text{(Open Epi version 3 Source calculator,)}$$

Where, Deff is the design effect, N is the finite population (360,000), p is the proportion (42.5%), d is the precision (0.05) and z is the z-value at the 95% confidence level (1.96).

The sample size was 751, calculated based on a prevalence of 42.5% of completion of T3 with a Population of 360,000 children under 5 years in the Volta and Oti regions, at the 95% confidence level, with a design effect of 2 and a precision of 5%. Adjusting for a non-response rate of 15%, we calculated the sample size to be 863.65 or approximately 864. However, 900 children were recruited. Also, a total of 69 clinicians were interviewed, thus one clinician from each facility.

## Sampling method

The sampling was done in two stages:

**Stage one (1): Selection of health facilities.** One district was randomly selected by ballot from each of the three ecological zones. All hospitals, health centres and CHPS compounds in the selected districts were included in the study. A total of 69 health facilities [(Hospitals (2), Health centres (40) and CHPS compounds (27)] from three Municipalities/Districts (Keta, Hohoe and Nkwanta) were used in the study.

**Stage two (2): Selection of participants.** One clinician from each of the selected health facilities was included in this study, but in the case where there was more than one clinician in a health facility, a YES or NO balloting was employed in the selection. Study participants (children under-five years with their parents/guardian) were selected using a convenient sampling technique (that is they were recruited as they exited the health facility after accessing services until the sample size was obtained). The sample was proportionately apportioned to each of the health facilities included in the study. To select participants from these facilities, the number of participants that were selected was based on the number of children under-five years seen at the OPD per facility. This proportionate sampling was based on the sample size calculated for this study. A child was eligible to participate in this study, if a clinician wrote in his/her folder that he/she had malaria.

## Data collection

The data was collected using a pre-tested semi-structured questionnaire. The Principal Investigator (PI) together with eight (8) other trained persons collected the data. Exit interviews were

conducted when the patients had been attended to and were leaving the health facility. Interviews were conducted in the English language or in the local dialect (Ewe) in the case where respondents did not understand or speak English. Information collected included the background characteristics of children, whether the child was tested before treatment was given, whether the child was given an ACT and was asked to come back for review.

The clinicians were interviewed using a semi-structured questionnaire. Information collected on the clinicians included their background, training received on malaria case management, and training on the T3 strategy as well as the challenges that confronted them in the implementation of the strategy.

## Definition of key terminologies

Clinician: Physician, Physician Assistant, State Registered Nurse, Enrolled Nurse or Community Health Officer who was assigned the roll of providing clinical care and was at post during the time of the study.

Test: Blood tested for malaria parasites using either microscopy or RDT.

Treat: Treatment given using approved ACTs (AS+AQ, AL or Dihydroartemisinin + Piperaquine (DHAP).

Track: Ask patient to return to the health facility for follow-up.

T3: Test, Treat and Track.

## Data management and analysis

Data collected were checked by the PI for completeness before entry into Epi data version 3.1 and was exported to STATA version 15.1 for analysis. The results were presented in tables and graphs. The Chi-square test and binary logistic regression were used to determine the association between completion of T3 strategy and such variables as demographic factors and health service-related factors. Challenges associated with the implementation of the T3 strategy were also presented.

## Ethical issues

Ethical clearance was obtained in May 2017 from the Ghana Health Service (GHS) Ethics Review Committee (ERC) (GHS-ERC) and the Dodowa Health Research Center (DHRC) Institutional Review Board (IRB) (DHRCIRB) with study approval numbers of GHS-ERC: 44/05/17 and DHRCIRB/25/05/17 respectively before the commencement of the study. Also, permission from the Municipal/District Health Directorates and the Health facilities was sought. A written informed consent was obtained from the (mother/guardian) of respondents as well as clinicians before commencement of the study.

## Results

### Distribution of background variables of children by level of health facility

A total of 900 children aged less than five years attending OPD with fever were surveyed from 69 health facilities comprising of 27 CHPS compounds, 40 health centres and 2 hospitals. Table 1 shows that, of the 900 children under 5 years enrolled, 314 (34.9%) were seen at the CHPS compound, 468 (52.0%) were seen at the health centre and 118 (13.1%) were seen at the hospital. The mean age of the children was 27.8 ± 15.6 months. Approximately half, 451 (50.1%) of the children were male.

A total of 69 clinicians were interviewed out of which 40 (58.0%) were from the health centre, 27 (39.1%) from CHPS compound and 2 (2.9%) from the Hospital. Over half, 39

**Table 1. Distribution of background characteristics of children under 5 years and clinicians by level of health facility.**

| Variable | CHPS Compound [N = 314] n (%) | Health Center [N = 468] n (%) | District Hospital [N = 118] n (%) | Total [N = 900] n (%) | Pearson Chi-square ($\chi^2$) | p-value |
|---|---|---|---|---|---|---|
| **Sex of child** | | | | | | |
| Male | 148 (47.1) | 240 (51.3) | 63 (53.4) | 451 (50.1) | | |
| Female | 166 (52.9) | 230 (48.7) | 55(46.6) | 449 (49.9) | 1.88 | 0.391 |
| **Mean age (in months) (SD)** | | | | 27.8 (15.55) | | |
| **Age groups (in months)** | | | | | | |
| <12 | 45 (14.3) | 90 (19.2) | 23 (19.5) | 158 (17.6) | | |
| 12–23 | 66 (21.0) | 107 (22.9) | 1.9 (16.1) | 192 (21.3) | | |
| 24–35 | 63 (20.1) | 94 (20.1) | 25 (21.2) | 182 (20.2) | | |
| 36–47 | 68 (21.7) | 92 (19.7) | 26 (22.0) | 186 (20.7) | | |
| 48–59 | 72 (22.9) | 85 (18.2) | 25 (21.2) | 182 (20.2) | 7.62 | 0.471 |
| **Background characteristics of clinicians** | | | | | | |
| | **N = 27** | **N = 40** | **N = 2** | **N = 69** | | |
| **Sex of clinician** | | | | | | |
| Male | 9 (33.3) | 20 (50.0) | 1 (50.0) | 30 (43.5) | | |
| Female | 18 (66.7) | 20 (50.0) | 1 (50.0) | 39 (56.5) | 1.86 | 0.395 |
| **Category of clinician** | | | | | | |
| CHO/CHN | 12 (44.4) | 13 (32.5)) | 0 (0.0) | 25 (36.2) | | |
| Enrolled/Diploma Nurse | 15 (55.6) | 16 (40.0) | 1 (50.0) | 32 (46.4) | | |
| Physician Assistant/ Doctor | 0 (0.0) | 11 (27.5) | 1 (50.0) | 12 (17.4) | 10.50 | 0.033 |
| **Awareness of T3** | | | | | | |
| Yes | 24 (88.9) | 33 (82.5) | 2 (100.0) | 59 (85.5) | 0.88 | 0.644 |
| No | 3 (11.1) | 7 (17.5) | 0 (0.0) | 10 (14.5) | | |
| **Receive formal training on how to use RDT** | | | | | | |
| Yes | 17 (63.0) | 19 (47.5) | 2 (100.0) | 38 (55.1) | | |
| No | 10 (37.0) | 21 (52.5) | 0 (0.0) | 31 (44.9) | 3.24 | 0.198 |
| **Received formal training on malaria case management** | | | | | | |
| Yes | 20 (74.1) | 25 (62.5) | 2 (100.0) | 47 (68.1) | | |
| No | 7 (25.9) | 15 (37.5) | 0 (0.0) | 22 (31.9) | 1.96 | 0.376 |
| **Availability of laboratory facility or easy access to laboratory** | | | | | | |
| Yes | 6 (22.2) | 21 (52.5) | 2 (100.0) | 29 (42.0) | | |
| No | 21 (77.8) | 19 (47.5) | 0 (0.0) | 40 (58.0) | 8.91 | 0.012 |
| **Able to use RDT** | | | | | | |
| Yes | 26 (96.3) | 40 (100.0) | 2 (100.0) | 68 (98.5) | | |
| No | 1 (3.7) | 0 (0.0) | 0 (0.0) | 1 (1.5) | 1.58 | 0.454 |

CHO/CHN = Community Health Officer/Community Health Nurse, RDT = Rapid Diagnostic Test, ACT = Artemisinin Combination Therapy T3 = Test, Treat, Track, N = Number sampled, n = number of responses

(56.5%) of the clinicians were female. The most common category of clinicians was Enrolled/Diploma nurses 32 (46.4%) followed by Community Health Officer (CHO)/ Community Health Nurse (CHN) 25 (36.2%) and the least were Physician Assistant/Doctors 12 (17.4%).

## Awareness and training on T3 strategy

Table 1 shows that out of the 69 clinicians, 59 (85.5%) of them were aware of the T3 strategy. Most, 68 (98.5%) of the clinicians knew how to conduct RDT, 47 (68.1%) had formal training on malaria case management and 38 (55.1%) had formal training on how to conduct RDT. Less than half, 29 (42.0%) of the clinicians said they had laboratory facilities available.

## Testing and treatment of malaria cases

Table 2 shows that, 818 (90.9%) of the 900 children enrolled were tested for malaria. The CHPS compound tested 296/314 (94.3%) children, health centres 418/468 (89.3%) and the hospitals 104/118 (88.1%). Overall, the prevalence of malaria among children tested was 600/818 (73.4%) with the highest prevalence occurring among children who attended CHPS compounds 251/296 (84.8%), health centres 294/418 (70.3%) and least was the hospitals 55/104 (52.9%). Of those who tested positive for malaria, 530/600 (88.3%) received treatment with ACTs of which 247/251 (98.4%) were treated at the CHPS Compounds, 230/294 (78.2%) from health centres and 53/55 (96.4%) at the hospitals. Also, 109/218 (50.0%) of children who tested

**Table 2. Distribution of blood test carried out, test results and treatment given by level of health facility.**

| Variable | CHPS Compound [N = 314] n (%) | Health Centre [N = 468] n (%) | Hospital [N 118] n(%) | Total [N = 900] n (%) | Pearson Chi-Square ($\chi^2$) | p-value |
|---|---|---|---|---|---|---|
| **Tested with RDT** | | | | | | |
| Yes | 296 (94.3) | 418 (89.3) | 104 (88.1) | 818 (90.9) | 6.81 | 0.033 |
| No | 18 (5.7) | 50 (10.7) | 14 (11.9) | 82 (9.1) | | |
| **Results of test** | **N = 296** | **N = 418** | **N = 104** | **N = 818** | | |
| Malaria positive | 251(84.8) | 294 (70.3) | 55 (52.9) | 600 (73.4) | 44.1 | <0.001 |
| **Test positive and treatment given** | | | | | | |
| | **N = 251** | **N = 294** | **N = 55** | **N = 600** | | |
| Positive Test & treated with ACT | 247 (98.4) | 230 (78.2) | 53 (96.4) | 530 (88.3) | | |
| Positive Test with no ACT treatment | 4 (1.6) | 64 (21.8) | 2 (3.6) | 70 (11.7) | 57.27 | <0.001 |
| **Test negative and treatment given** | | | | | | |
| | **N = 45** | **N = 124** | **N = 49** | **N = 218** | | |
| Negative Test & treated with ACT | 21 (46.7) | 69 (55.6) | 19 (38.8) | 109 (50.0) | | |
| Negative Test with no ACT treatment | 24 (53.3) | 55 (44.4) | 30 (61.2) | 109 (50.0) | 4.25 | 0.119 |
| **No test and treatment given** | | | | | | |
| | **N = 18** | **N = 50** | **N = 14** | **N = 82** | | |
| No Test but treated with ACT | 16 (88.9) | 39 (78.0) | 12 (85.7) | 67 (81.7) | | |
| No Test and no treatment with ACT | 2 (11.1) | 11 (22.0) | 2 (14.3) | 15 (18.3) | 1.23 | 0.540 |
| **Asked to bring child back for review** | | | | | | |
| **Yes** | 221 (70.4) | 256 (54.7) | 39 (33.1) | 516 (57.3) | | |
| **No** | 93 (29.6) | 212 (45.3) | 79 (66.9) | 384 (42.7) | 51.62 | <0.001 |
| **Complete T3 by health facility** | | | | | | |
| Not complete T3 | 112 (35.7) | 287 (61.3) | 93 (78.8) | 492 (54.7) | | |
| Completed T3 | 202 (64.3) | 181 (38.7) | 25 (21.2) | 408 (45.3) | 81.86 | <0.001 |

RDT = Rapid Diagnostic Test, ACT = Artemisinin Combination Therapy, T3 = Test, Treat, Track, N = Number sampled, n = number of responses

negative for malaria also were treated with ACTs. Again, 67/82 (81.7%) of children who were not tested for malaria were also treated with ACTs.

### Tracking of malaria cases

Overall, 516/900 (57.3%) of the children were asked to return to the health facility for review (Table 2), with the highest proportion being those attending the CHPS compound, 221/314 (70.4%) (95% CI: 65.0%–75.4%) asked to return, health centre, 256/468 (54.7%) (95% CI: 50.1%–59.3%) and hospital, 39/118 (33.1%) (95% CI: 24.7%–42.3%).

### Completion of T3 by health facility

Table 2 shows that, overall, 408/900 (45.3%) completed the T3 strategy and the highest completion proportion was the CHPS compounds, 202/314 (64.3%) (95% CI: 58.8%-69.6%) followed by the health centres, 181/468 (38.7%) (95% CI: 34.2%-43.3%) and the hospitals, 25/118 (21.2%) (95% CI: 14.2%-29.7%).

### Associations between background characteristics of children, clinicians and the odds of completing T3

Our results in the multivariable regression analysis found that health centres and CHPS compounds were 2.39 and 6.79 times more likely to complete T3 as compared to hospitals [AOR = 2.39 (95% CI: 1.48, 3.87), p<0.001] and [AOR = 6.79 (95% CI: 4.12, 11.21), p<0.001] respectively. Clinicians who had training on the use of RDT to conduct malaria test were 1.48 times more likely to complete T3 [AOR = 1.48 (95% CI: 1.10, 2.00), p = 0.010]. Similarly, clinicians who had access to laboratory facilities were 2.08 times more likely to complete T3 [AOR = 2.08 (95% CI: 1.55, 2.79), p<0.001] (Table 3).

### Challenges confronting the T3 strategy

Clinicians' negative perception that RDTs do not give accurate results 46 (66.7%), lack of diagnostic facilities 45 (65.2%), frequent RDT stock outs 41 (59.4%) and inadequate training on use of RDTs/malaria case management 40 (58.0%) were main challenges mentioned. Frequent ACTs stock outs 28 (40.6%) and patients resisting diagnostic testing 20 (29.0%) were also mentioned as challenges.

## Discussion

The WHO's T3 strategy for malaria control recommends that every suspected malaria case should be tested prior to treatment with an antimalarial drug [6]. The testing could either be done by using malaria rapid diagnostic test (RDT) or microscopy. This recommendation by the WHO seeks to put an end to the presumptive treatment of malaria which leads to drug wastage and under-treatment of other febrile illnesses and development of drug resistance. In addition, all malaria cases should be tracked in a surveillance system to be able to measure malaria prevalence and impact of malaria interventions to guide policy and practice [7].

### Proportion of suspected malaria cases tested by clinicians before treatment

In theory, the availability of reliable easy-to-use tests such as RDTs should result in a switch from presumptive treatment based on signs and symptoms alone to parasite-based diagnosis and treatment based on test results [19]. Our study found a high overall test rate of 90.9% even though we did not achieve the 100% target set by WHO. However, our finding is far higher than the 38.8% reported by Nyandigisi and colleagues from Kenya, 26.4% by Osei-Kwakye

**Table 3. Association between socio-demographic characteristics and other risk factors of study participants and the odds of completing T3.**

| Characteristics | COR (95%CI) p-value | AOR (95%CI) p-value |
|---|---|---|
| **Sex of children** | | |
| Male | Ref. | Ref. |
| Female | 1.21 (0.92, 1.57) 0.162 | 1.17 (0.88, 1.54) 0.276 |
| **Age group of children (in months)** | | |
| <12 | Ref. | Ref. |
| 12–23 | 0.91(0.59, 1.39) 0.649 | 0.81 (0.52, 1.26) 0.348 |
| 24–35 | 1.32 (0.86, 2.03) 0.200 | 1.26 (0.80, 1.98) 0.312 |
| 36–47 | 1.07(0.70, 1.64) 0.768 | 0.97 (0.62, 1.52) 0.903 |
| 48–59 | 1.24 (0.81, 1.90) 0.327 | 1.12 (0.71, 1.76) 0.619 |
| **Health Facility** | | |
| Hospital | Ref. | Ref. |
| Health Centre | 2.35 (1.45, 3.79) <0.001 | **2.39(1.48, 3.87) <0.001** |
| CHPS Compound | 6.71(4.08, 11.04) <0.001 | **6.79(4.12, 11.21) <0.001** |
| **Clinician's Sex** | | |
| Male | Ref. | Ref. |
| Female | 0.75 (0.58, 0.98) 0.032 | 0.96 (0.73, 1.47) 0.844 |
| **Category of clinician** | | |
| CHO/CHN | Ref. | Ref. |
| Enrolled/Diploma nurse | 1.32 (0.95, 1.83) 0.099 | 1.04 (0.73, 1.47) 0.844 |
| Physician Assistant/Doctor | 1.77 (1.24, 2.54) 0.002 | 1.33 (0.91, 1.94) 0.146 |
| **Availability of laboratory** | | |
| No | Ref. | Ref. |
| Yes | 2.35 (1.79, 3.07) <0.001 | **2.08 (1.55, 2.79) <0.001** |
| **Training on use of RDT** | | |
| No | Ref. | Ref. |
| Yes | 1.80 (1.35, 2.39) <0.001 | 1.48 (1.10, 2.00) 0.010 |

COR = Crude Odds Ratio, AOR = Adjusted Odds Ratio, RDT = Rapid Diagnostic Test, CHPS = Community-Based Health Planning and Services, CHO = Community Health Officer, CHN = Community Health Nurse

et al., from Kintampo, Ghana and 43.8% by Abdelgader et al., in the Republic of Sudan [11, 20, 21]. This could be due to the fact that, the T3 strategy was launched in 2012 and the implementation started in 2013 however, these studies were conducted before the commencement of the T3 strategy implementation.

A similar study in the Ho Municipality of Ghana by Kankpetinge et al., reported a test rate of 58.5% [13], lower than the findings of this current study. The reason for the lower test rate could be that their study was conducted in May 2015 whilst this current study was conducted in September 2017, three and five years respectively after the commencement of the implementation of T3 strategy. According to the Ghana Health Service (GHS) annual report 2016 there was massive training of clinicians on the T3 strategy in Ghana in 2015 [22]. The higher test rate in this current study could be attributed to the improvement in clinician's adherence to the T3 strategy as a result of the training.

Our study found the prevalence of malaria among children tested to be 73.4%. This is higher than what was reported in the Republic of Sudan [21] and in Ho, Ghana [13] who reported the prevalence of malaria to be 37.7% and 52.8% respectively. The high prevalence of

malaria in our study could be attributed to the fact that, majority of the children with fever were tested and therefore malaria cases could not have been missed. In addition, the time data was collected could be a contributory factor to the prevalence of malaria since data was collected in September which the high transmission season.

### Confirmed malaria cases receiving treatment according to the T3 policy

In this current study, a large proportion (88.3%) of the children who tested positive for malaria were treated with ACTs, however this is slightly lower than what was reported by Agandaa et al., (91.2%) in Bongo district [12] and 100% by Kankpetinge et al., in the Ho Municipality [13]. The reason for a lower treatment in this current study could be attributed to frequent ACTs stock outs, (40.6%) which was reported by clinicians as a challenge. The current study found that half, (50.0%) of the children who tested negative for malaria were also treated with ACT compared to 21.9% and 17% reported in other studies [13, 21]. The high treatment of negative test patients with ACT in this current study could be attributed to the fact that most, (66.7%) clinicians believed RDTs do not give accurate results. Also, in this study, 81.7% of those who were not tested were also treated with ACTs compared to whilst 52.0% and 10.2% reported in other studies [13, 21]. The high treatment rate of non-tested case with ACT in this current study could be attributed to the fact that there was a high rate (59.4%) of RDT stock out reported by clinicians.

Despite the fact that the expected target for treatment with ACT based on positive test results is 100%, the observation made in the Volta and Oti regions is an indication of an improvement which is in line with the observations made by Agandaa et al., in the Bongo district of Ghana (2016) Kankpetinge et al., in the Ho Municipality of Ghana (2015), Zurovac et al., in Kenya (2014) and Udoh et al., in Nigeria (2013) [12, 13, 23, 24]. Nonetheless, it is evident that the few cases presumptively treated must be re-examined to improve the performance of the T3 strategy of malaria control among children under-five in the regions and also early detect other conditions which may be responsible for febrile presentation of these children.

### Tracking of children after treatment

The reason for tracking was that once malaria cases are being asked to come for review, cure rates and cure status of cases could be ascertained. Again, cases still harboring the malaria parasites after treatment could be duly managed and could also lead to the elimination of the parasites. Thiam et al., from Senegal found that the ability to track the confirmed cases and track the impact of antimalarial interventions through the widespread use of parasite-based diagnosis will enable malaria elimination to be seriously considered [25]. In this current study, more than half (57.3%) of the children tested were asked by clinicians to return for review compared to 38.5% reported by in Ho, Ghana [13].

Further, in this current study, CHPS compounds recorded the highest tracking rate (70.4%) compared to other facilities.. This could be attributed to the fact that CHPS compounds are in close proximity to the people than hospitals. This finding was comparable to tracking rate of 70.6% reported in Bongo district, Ghana for CHPS compounds compared to other health facilities [12].

### Completion of T3

A good performance of the T3 strategy for malaria is measured by being tested, treated with an antimalarial and asked to come for review. Thus, all the 3 Ts must be in place. Missing a "T" means the strategy or policy was not well adhered to. This current study found that 45.3% of the children completed all the 3 Ts. This is closer to what was reported (42.5%) in the Bongo District of Ghana [12]. This can be attributed to the low tracking of cases where close to half of

the children were not asked to come for review. This study found that CHPS compounds and health centres were 6.79 and 2.39 times more likely to complete T3 compared to hospitals which is due to the low tracking of cases in the hospitals.

## Challenges associated with the implementation of the T3 strategy

The main challenge stated by clinicians to the successful implementation of the T3 policy was clinicians' negative perception that RDTs do not give accurate results (66.7%) contrary to findings of Agandaa et al., in Bongo and Kankpetinge et al., in Ho where frequent RDT stock out was the major challenge faced by clinicians [12, 13]. Studies in Burkina Faso have shown that clinicians do not trust the results of RDTs as reported in our study [26, 27]. In Ghana the RDT kit mainly supplied to health facilities by the National Malaria Control Programme is the SD BIOLINE Malaria Ag P.F (HRP2/pLDH) manufactured by the Abbott Diagnostics used for the rapid qualitative detection of Histidine-rich Protein II (HRP-II) antigen and Lactose dehydrogenase (pLDH) from malaria Plasmodium falciparum. Studies have shown that this test kit has relatively good sensitivity/specificity compared to traditional testing methods like microscopy and polymerase chain reaction (PCR) [28, 29]. Therefore, clinicians need to test with RDTs when available instead of assuming they don't give accurate results.

Although data for this study was collected about 5 years ago, the findings are still relevant in informing interventions to improve adherence to the T3 strategy in health facilities in Ghana since to the best of our knowledge no specific intervention has been implemented in the regions the study was conducted to improve adherence to the strategy.

## Limitations of the study

The limitation of the study was that, data could not be collected to determine seasonal trends hence the effects of seasonal variations of the incidence of malaria could not be determine because the study employed a cross-sectional study design. Even though exit interviews were conducted to collect data from the participants, there was the possibility of recall bias because their responses might have been influenced by the conditions of their children.

## Conclusion

There has been improvement in testing before treatment and a reduction in presumptive treatment for malaria in the Volta and Oti regions of Ghana. Adherence to treatment with ACTs based on test results was very high, tracking of children tested and treated was moderate (57.3%) and completing T3 was low. Key predictors to completion of T3 were availability of laboratory services, training of clinicians to use RDTs, working at lower-level facilities such as CHPS compound and health centre. Clinicians' negative perception that RDTs do not give accurate results is a challenge and this can affect the T3 strategy for malaria control.

## Recommendations

In order to improve upon completeness of the T3 strategy and sustain the high coverage of testing and treatment with ACTs, the National Malaria Control Program (NMCP) in collaboration with the Regional Health Directorate and Medical store should ensure regular supply of ACTs and RDT to address the frequent ACT and RDT stock out. Clinicians should be trained and given periodic refresher trainings to test all fever cases with RTDs and not to presume they do not give accurate results. The GHS should ensure that health facilities are equipped with diagnostic facilities to increase compliance to T3 strategy. Targeted intervention in the form of monitoring and supervision, retraining of staff to address the differences in testing,

treating and tracking of malaria cases at facility level should be implemented. There is the need to explore effectiveness of other means such as contacting patients by telephone or text messages or using contact persons in the communities to track malaria patients.

## Supporting information

**S1 Dataset. Minimal dataset underlying the results presented in this study.**
(XLSX)

## Acknowledgments

We are grateful to Dr Felix Doe, Dr Laud Boateng and Miss Titiati and the staff of the Keta, Hohoe and Nkwanta South Municipal Health Directorate. We would like to thank the interviewers as well as the mothers and children who participated in the study.

## Author Contributions

**Conceptualization:** Margaret Kweku, William K. Blankson, Haruna M. Salisu, Francis Arizie, Fortress Y. Aku, Martin Adjuik.

**Formal analysis:** Margaret Kweku, Joyce B. Der, William K. Blankson, Haruna M. Salisu, Francis Arizie, Sorengmen A. Ziema, Fortress Y. Aku, Martin Adjuik.

**Investigation:** Margaret Kweku, William K. Blankson, Haruna M. Salisu, Francis Arizie.

**Methodology:** Margaret Kweku, Joyce B. Der, William K. Blankson, Haruna M. Salisu, Francis Arizie, Fortress Y. Aku, Martin Adjuik.

**Project administration:** Margaret Kweku.

**Supervision:** Margaret Kweku, Martin Adjuik.

**Validation:** Margaret Kweku, Martin Adjuik.

**Visualization:** Margaret Kweku.

**Writing – original draft:** Margaret Kweku, Joyce B. Der, William K. Blankson, Haruna M. Salisu, Francis Arizie, Sorengmen A. Ziema, Jonathan M. Gmanyami, Fortress Y. Aku, Martin Adjuik.

**Writing – review & editing:** Margaret Kweku, Joyce B. Der, William K. Blankson, Haruna M. Salisu, Francis Arizie, Sorengmen A. Ziema, Jonathan M. Gmanyami, Fortress Y. Aku, Martin Adjuik.

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
