## [Decision Letter · Decision Letter 0]

18 Jul 2022

PONE-D-21-40765Assessment of the performance and challenges in the implementation of the test, treat and track (T3) strategy for malaria control among children under-five years in GhanaPLOS ONE

Dear Dr. Der,

Thank you for submitting your manuscript to PLOS ONE. After careful consideration, we feel that it has merit but does not fully meet PLOS ONE’s publication criteria as it currently stands. Therefore, we invite you to submit a revised version of the manuscript that addresses the points raised during the review process. You are requested to address all points raised by the reviewers. it is also noted that samples were collected a considerable time ago. Please address this issue in your discussion. Also elaborate on the current validity of your observations

We look forward to receiving your revised manuscript.

Kind regards,

Henk D. F. H. Schallig, Ph.D

Academic Editor

PLOS ONE

Journal Requirements:

2. . In your Data Availability statement, you have not specified where the minimal data set underlying the results described in your manuscript can be found. PLOS defines a study's minimal data set as the underlying data used to reach the conclusions drawn in the manuscript and any additional data required to replicate the reported study findings in their entirety. All PLOS journals require that the minimal data set be made fully available. For more information about our data policy, please see http://journals.plos.org/plosone/s/data-availability.

Reviewers' comments:

Reviewer's Responses to Questions

**Comments to the Author**

1. Is the manuscript technically sound, and do the data support the conclusions?

Reviewer #1: Yes

Reviewer #2: No

2. Has the statistical analysis been performed appropriately and rigorously? 

Reviewer #1: Yes

Reviewer #2: I Don't Know

3. Have the authors made all data underlying the findings in their manuscript fully available?

Reviewer #1: Yes

Reviewer #2: Yes

4. Is the manuscript presented in an intelligible fashion and written in standard English?

Reviewer #1: Yes

Reviewer #2: No

5. Review Comments to the Author

Reviewer #1: This paper addresses an important issue in the implementation of T3 startegy of WHO for malaria. For me the most important finding was that Health Workers do not trust the RDTS. This has also been found in previous studies, for example :

1) The effect of malaria rapid diagnostic tests results on antimicrobial prescription practices of health care workers in Burkina Faso. Bonko MDA, et al. Ann Clin Microbiol Antimicrob. 2019 Jan 28;18(1):5. doi: 10.1186/s12941-019-0304-2

2) Accuracy of a Plasmodium falciparum specific histidine-rich protein 2 rapid diagnostic test in the context of the presence of non-malaria fevers, prior anti-malarial use and seasonal malaria transmission.

Kiemde F, et al Malar J. 2017 Jul 20;16(1):294. doi: 10.1186/s12936-017-1941-6

And these papers (or comparable ones should be mentioned in the introduction or discussion.

It is noted that the data were collected about 5 years ago. How valid are these nowadays? Why did it take so long to prepare a manuscript. Please explain.

It would be very useful to discuss the type of RDT(s) that have been used in the faciities for the diagnosis of malaria. Are these notorious in providing incorrect diagnosis?

Also elaborate on ACTs (and other treatments provided.

The figures are redundant and can be sumarised in the main body of the text

Reviewer #2: General comments:

As the title of the manuscript describe, the authors have assessed the performance and challenges in the implementation a strategy called T3 (test-treat-track) for the control of malaria among children under 5 years in Ghana.

No general comment to do.

Specific comments:

Abstract: Results:

Line39-40: “Most children, 818/900 (90.9%) were tested for malaria and 600/818 (73.4%) were positive for malaria parasitaemia.” Which test was used detect malaria parasitemia? RDT or microscopy? Authors need to specify.

Line 43: “ hildren completed T3 with CHPS compound “. Please give the explanation of this abbreviation.

Introduction

Line 63: “challenges of T3 strategy”. It the first time to use T3 in the main text. Please explain it.

General comment: 3 page for the introduction is too long. I will ask authors to summary the introduction 1 page and half maximum. Some paragraphs can be summarized in few lines to make it consistent. I encourage the authors to review the introduction.

Materials and methods

Study site description

Line 128-134 “the region are two ………. mountainous”. This section is not relevant for the reader. Maybe use a map to show the study site.

Line 137-139: Not relevant. Reader want information that could be helpful for the reader to understand your study and data.

General comment: There are more information that are not relevant for the ready. I will suggest the authors to summarize and give the relevant ones.

Study population

Line 146: “The second study “. Is it a cross-section survey or another study different to the main study? I think after enrolment of febrile children, cross-sectional survey were made to collect qualitative information. This should be clarify.

Study design

I note that no biological data has been collected. Only interview of healthcare workers or care givers.

Results

Prefer to not make comment.

Discussion

The discussion is made in 4 pages and half. In addition, In addition, the study design is not strong to make a solid analysis. Data collected are main based on information collected from care givers and healthcare workers. This has been noted by the authors and I think this is a major bias.

For me this paper cannot be accepted for publication.

6. PLOS authors have the option to publish the peer review history of their article (what does this mean?). If published, this will include your full peer review and any attached files.

Reviewer #1: No

Reviewer #2: No

---

## [Author Response · Author response to Decision Letter 0]

31 Aug 2022

Reviewer 1 

Comment: This paper addresses an important issue in the implementation of T3 strategy of WHO for malaria. For me the most important finding was that Health Workers do not trust the RDTS. This has also been found in previous studies, for example :

1) The effect of malaria rapid diagnostic tests results on antimicrobial prescription practices of health care workers in Burkina Faso. Bonko MDA, et al. Ann Clin Microbiol Antimicrob. 2019 Jan 28;18(1):5. doi: 10.1186/s12941-019-0304-2

2) Accuracy of a Plasmodium falciparum specific histidine-rich protein 2 rapid diagnostic test in the context of the presence of non-malaria fevers, prior anti-malarial use and seasonal malaria transmission.

Kiemde F, et al Malar J. 2017 Jul 20;16(1):294. doi: 10.1186/s12936-017-1941-6

And these papers (or comparable ones should be mentioned in the introduction or discussion.

Response: Thank you for your comment and for suggesting these two important papers that had a similar challenge of clinicians not trusting the results of RDTs as found in our study. 

As suggested we have now mentioned these two papers in our discussion. This can be found on lines 448-449 and reads as “Studies in Burkina Faso have shown that clinicians do not trust the results of RDTs as reported in our study [26, 27]”

Comment: It is noted that the data were collected about 5 years ago. How valid are these nowadays? Why did it take so long to prepare a manuscript. Please explain.

Response: Thank you for your comment. We acknowledge that the data were collected 5 years ago but we believe that the findings are still valid since no major intervention has been implemented by the Ghana Malaria Control Programme to address the gaps identified in the regions the study was conducted.

The reason for which it took so long for the manuscript to be prepared was because the data collected was to be used for a manuscript writing workshop which was to be used as a learning process for students of the School of Public Health of the University of Health and Allied Sciences where the Principal Investigator is a Professor. However, the manuscript writing workshop could not come off due to some interruptions in the School’s academic calendar which was further worsened by the COVID 19 pandemic and Schools were closed down. The workshop could not be organized via zoom since most of these students lived in remote areas where internet connectivity was poor hence the delay in the manuscript preparation. 

We have now included this in our discussion and it can be found on lines 459-462 and reads “Although data for this study was collected about 5 years ago, the findings are still relevant in informing interventions to improve adherence to the T3 strategy in health facilities in Ghana since to the best of our knowledge no specific intervention has been implemented in the regions the study was conducted to improve adherence to the strategy.”

Comment: It would be very useful to discuss the type of RDT(s) that have been used in the facilities for the diagnosis of malaria. Are these notorious in providing incorrect diagnosis?

Response: Thank you for your comment. We have now included in our discussion the type of RDT kits used in the health facilities and mentioned studies that have evaluated this test kit and found it to have satisfactory agreement with traditional methods of testing like microscopy and PCR.

This revision is found on lines 449-455 and reads “In Ghana the RDT kit mainly supplied to health facilities by the National Malaria Control Programme is the SD BIOLINE Malaria Ag P.F (HRP2/pLDH) manufactured by the Abbott Diagnostics used for the rapid qualitative detection of Histidine-rich Protein II (HRP-II) antigen and Lactose dehydrogenase (pLDH) from malaria Plasmodium falciparum. Studies have shown that this test kit has relatively good sensitivity/specificity compared to traditional testing methods like microscopy and polymerase chain reaction (PCR) [28, 29]. Therefore, clinicians need to test with RDTs when available instead of assuming they don’t give accurate results”

Comment: Also elaborate on ACTs (and other treatments provided. 

Response: We have revised the background to include information on ACTs used in health facilities in Ghana according to the national malaria treatment guidelines. This can be found on lines 90-97.

Comment: The figures are redundant and can be sumarised in the main body of the text 

Response: Thank you for the comment. We have removed the figures and summarized them in the main text.

Reviewer 2 

Comment: As the title of the manuscript describe, the authors have assessed the performance and challenges in the implementation a strategy called T3 (test-treat-track) for the control of malaria among children under 5 years in Ghana.

No general comment to do. 

Response: Thank you for your brief overview of the manuscript

Specific comments:

Comment: Abstract: Results:Line39-40: “Most children, 818/900 (90.9%) were tested for malaria and 600/818 (73.4%) were positive for malaria parasitaemia.” Which test was used detect malaria parasitemia? RDT or microscopy? Authors need to specify. 

Response: Thank you for your comment. We have revised the results in the abstract to indicate the test that was used. This can be found on line 38-39 and reads as “Most children, 818/900 (90.9%) were tested for malaria and 600/818 (73.4%) were positive for malaria parasitaemia using rapid diagnostic test”

Comment: Line 43: “ Children completed T3 with CHPS compound “. Please give the explanation of this abbreviation. 

Response: We have written this term now in full and the sentence now reads as “Only 408/900 (45.3%) children completed T3 with Community Health-based Planning Services (CHPS) compound having the highest completion rate 202/314 (64.3%)” This can be found on line 42.

Introduction

Comment: Line 63: “challenges of T3 strategy”. It the first time to use T3 in the main text. Please explain it. 

Response: Thank you for the comment. Based on another comment, where you suggested that we revise the introduction to make it shorter, we have deleted the portion that had this statement.

Comment: General comment: 3 page for the introduction is too long. I will ask authors to summary the introduction 1 page and half maximum. Some paragraphs can be summarized in few lines to make it consistent. I encourage the authors to review the introduction.

Response: Thank you for your comment. We have revised the introduction and reduced it significantly. 

Materials and methods

Comment: Study site description

Line 128-134 “the region are two ………. mountainous”. This section is not relevant for the reader. Maybe use a map to show the study site

Line 137-139: Not relevant. Reader want information that could be helpful for the reader to understand your study and data.

General comment: There are more information that are not relevant for the ready. I will suggest the authors to summarize and give the relevant ones. 

Response: Thank you for your comment. We have revised the study area to take out most of the descriptive information and used a map to show the municipalities/districts in which the study was conducted. The map is labelled as Fig 1.

Comment: Study population

Line 146: “The second study “. Is it a cross-section survey or another study different to the main study? I think after enrolment of febrile children, cross-sectional survey were made to collect qualitative information. This should be clarify. 

Response: The study was a cross-sectional study that involved two different populations. The first population was caregivers of children under 5 years with a history of fever who were interviewed to find out if they had been tested for and or treated for malaria. The second population was clinicians assigned the duties of prescribing during the study period and were available. They were interviewed to determine the challenges hindering the implementation of the T3 strategy in health facilities.

The study population in the manuscript has been revised to make it clearer. This can be found on lines 169-172 and reads “The study had two different populations: (i) children under five years with their caregivers who reported to the health facilities with fever or history of fever and were managed by clinicians and (ii) clinicians rendering services at the various health facilities during the time of the study”

Comment: Study design

I note that no biological data has been collected. Only interview of healthcare workers or care givers. 

Response: Thank you for this observation. One of the objectives of the study was to assess the implementation of the T3 strategy as part of routine care in health facilities. That is why we interviewed care givers of children under 5 years who had accessed routine care at the health facilities to find out if the T3 strategy was followed. The use of exit interviews made it difficult to collect biological data such as verifying the test result written in the folder of the child with what was recorded in the OPD or laboratory register. The other objective was to determine the challenges faced by clinicians in the implementation of the T3 strategy in health facilities, hence there was no need for biological data to answer this objective. 

Comment: Results

Prefer to not make comment. 

Response: Thank you

Comment: Discussion

The discussion is made in 4 pages and half. In addition, In addition, the study design is not strong to make a solid analysis. Data collected are main based on information collected from care givers and healthcare workers. This has been noted by the authors and I think this is a major bias. 

Response: Thank you for the comment. We have revised the discussion to shorten the length.

However, we politely disagree with the author that our study design was not strong to make solid analysis. The study design used enabled us to answer all the objectives of the study and our findings are relevant to the Ghana National Malaria Control Programme in identifying aspects of the T3 strategy that needs improvement. 

We did state in our limitations that the use of cross-sectional study design made it impossible to determine seasonal variations of malaria but findings from this study are very relevant as stated above.

Comment: For me this paper cannot be accepted for publication. 

Response: Thank you, we appreciate your comment but we believe the revisions we have made to the manuscript based on the valuable comments you have given has improved the quality of the manuscript and it can be considered for publication

Academic Editor 

It is also noted that samples were collected a considerable time ago. Please address this issue in your discussion. Also elaborate on the current validity of your observations.

Response: Thank you for your comment. We agree that the data was collected 5 years ago. In response to an earlier comment by Reviewer 1 above, we have explained this point and included it in the discussion section of the revised manuscript.

This can be found on lines 459-462 and reads “Although data for this study was collected about 5 years ago, the findings are still relevant in informing interventions to improve adherence to the T3 strategy in health facilities in Ghana since to the best of our knowledge no specific intervention has been implemented in the regions the study was conducted to improve adherence to the strategy.”

---

## [Decision Letter · Decision Letter 1]

28 Sep 2022

PONE-D-21-40765R1Assessment of the performance and challenges in the implementation of the test, treat and track (T3) strategy for malaria control among children under-five years in GhanaPLOS ONE

Dear Dr. Der,

Thank you for submitting your manuscript to PLOS ONE. After careful consideration, we feel that it has merit but does not fully meet PLOS ONE’s publication criteria as it currently stands. Therefore, we invite you to submit a revised version of the manuscript that addresses the points raised during the review process. The authors must address (also in line with PLOS ONE policy) issues related to access to data (preferably open access) and ethics (statement, date of review, date of approval, protocol number, reviewing board etc.). This is important to include otherwise we can not further accpet the paper. Please submit your revised manuscript by Nov 12 2022 11:59PM. If you will need more time than this to complete your revisions, please reply to this message or contact the journal office at plosone@plos.org. Please include the following items when submitting your revised manuscript:A rebuttal letter that responds to each point raised by the academic editor and reviewer(s). You should upload this letter as a separate file labeled 'Response to Reviewers'.A marked-up copy of your manuscript that highlights changes made to the original version. You should upload this as a separate file labeled 'Revised Manuscript with Track Changes'.An unmarked version of your revised paper without tracked changes. You should upload this as a separate file labeled 'Manuscript'.If applicable, we recommend that you deposit your laboratory protocols in protocols.io to enhance the reproducibility of your results. Protocols.io assigns your protocol its own identifier (DOI) so that it can be cited independently in the future. For instructions see: https://journals.plos.org/plosone/s/submission-guidelines#loc-laboratory-protocols. Additionally, PLOS ONE offers an option for publishing peer-reviewed Lab Protocol articles, which describe protocols hosted on protocols.io. Read more information on sharing protocols at https://plos.org/protocols?utm_medium=editorial-email&utm_source=authorletters&utm_campaign=protocols.

We look forward to receiving your revised manuscript.

Kind regards,

Henk Schallig, Ph.D

Academic Editor

PLOS ONE

Journal Requirements:

Reviewers' comments:

Reviewer's Responses to Questions

**Comments to the Author**

1. If the authors have adequately addressed your comments raised in a previous round of review and you feel that this manuscript is now acceptable for publication, you may indicate that here to bypass the “Comments to the Author” section, enter your conflict of interest statement in the “Confidential to Editor” section, and submit your "Accept" recommendation.

Reviewer #1: All comments have been addressed

Reviewer #2: All comments have been addressed

2. Is the manuscript technically sound, and do the data support the conclusions?

Reviewer #1: Yes

Reviewer #2: Partly

3. Has the statistical analysis been performed appropriately and rigorously? 

Reviewer #1: I Don't Know

Reviewer #2: I Don't Know

4. Have the authors made all data underlying the findings in their manuscript fully available?

Reviewer #1: No

Reviewer #2: Yes

5. Is the manuscript presented in an intelligible fashion and written in standard English?

Reviewer #1: Yes

Reviewer #2: Yes

6. Review Comments to the Author

Reviewer #1: I think that the authors have suffiicently addressed my concerns. However, to be in line with PLOS ONE policy, there must be statements on access to data (now only a minimal data set has been made available) and ethics should be addressed.

Reviewer #2: (No Response)

7. PLOS authors have the option to publish the peer review history of their article (what does this mean?). If published, this will include your full peer review and any attached files.

Reviewer #1: No

Reviewer #2: No

---

## [Author Response · Author response to Decision Letter 1]

12 Nov 2022

Academic Editor

Comment: Please review your reference list to ensure that it is complete and correct. If you have cited papers that have been retracted, please include the rationale for doing so in the manuscript text, or remove these references and replace them with relevant current references. Any changes to the reference list should be mentioned in the rebuttal letter that accompanies your revised manuscript. If you need to cite a retracted article, indicate the article’s retracted status in the References list and also include a citation and full reference for the retraction notice.

Response: Thank you for the comment. We have reviewed our reference list and updated reference 8 which was incomplete. We have also updated the URL for reference 15 which initially was inaccurate. These changes can be found on page 25, line 450 and page 26, line 479. We found none of our papers cited to have been retracted. 

Reviewer 1

Comment: I think that the authors have sufficiently addressed my concerns. However, to be in line with PLOS ONE policy, there must be statements on access to data (now only a minimal data set has been made available) and ethics should be addressed.

Response: Thank you accepting the revisions we have made based on the comments you provided to improve the manuscript. 

Access to data: we have provided the minimum dataset as a supporting information which is one of the acceptable data sharing methods for Plos One as per their data availability requirements. The data set provided contains all that is required to replicate all the study findings reported in the manuscript. We added a data availability statement under the data availability section of the online submission portal and the statement reads as “All relevant data are within the manuscript and its supporting information files”.

Although the preferred option is to deposit the data in a data repository, unfortunately our institution has no data repository for us to deposit our dataset. Also, this study did not receive any funding so we are unable to pay for the data to be deposited in a public data repository.

Ethics: we have a sub section titled ethical issues under the methods section of our manuscript where we have stated the ethics review boards that approved the study and the ethics approval numbers. We have revised the section to include the date of review and it reads as “Ethical clearance was obtained in May 2017 from the Ghana Health Service (GHS) Ethics Review Committee (ERC) (GHS-ERC) and the Dodowa Health Research Center (DHRC) Institutional Review Board (IRB) (DHRCIRB) with study approval numbers of GHS-ERC: 44/05/17 and DHRCIRB/25/05/17 respectively before the commencement of the study. Also, permission from the Municipal/District Health Directorates and the Health facilities was sought. A written informed consent was obtained from the (mother/guardian) of respondents as well as clinicians before commencement of the study”. This can be found on page 11, lines 216-223 of the revised manuscript.

Reviewer 2

Comment: No comment

Response: Thank you for your comments that helped improve the manuscript.

---

## [Editor Report · Decision Letter 2]

21 Nov 2022

Assessment of the performance and challenges in the implementation of the test, treat and track (T3) strategy for malaria control among children under-five years in Ghana

PONE-D-21-40765R2

Dear Dr. Der,

We’re pleased to inform you that your manuscript has been judged scientifically suitable for publication and will be formally accepted for publication once it meets all outstanding technical requirements.

Kind regards,

Henk Schallig, Ph.D

Academic Editor

PLOS ONE
---

## [Editor Report · Acceptance letter]

28 Nov 2022

PONE-D-21-40765R2 

Assessment of the performance and challenges in the implementation of the test, treat and track (T3) strategy for malaria control among children under-five years in Ghana 

Dear Dr. Der:

I'm pleased to inform you that your manuscript has been deemed suitable for publication in PLOS ONE. Congratulations! Your manuscript is now with our production department. 

Kind regards, 

on behalf of

Dr. Henk Schallig 

Academic Editor

PLOS ONE